# Application of Improved Particle Swarm Optimization for Navigation of Unmanned Surface Vehicles

**DOI:** 10.3390/s19143096

**Published:** 2019-07-13

**Authors:** Junfeng Xin, Shixin Li, Jinlu Sheng, Yongbo Zhang, Ying Cui

**Affiliations:** 1College of Electromechanical Engineering, Qingdao University of Science and Technology, Qingdao 266061, China; 2Transport College, Chongqing Jiaotong University, Chongqing 400074, China; 3Qingdao National Marine Science Research Center, Qingdao 266071, China

**Keywords:** travelling salesman problem, particle swarm optimization, parameter setting, random grouping inversion, unmanned surface vehicle, multi-sensor data

## Abstract

Multi-sensor fusion for unmanned surface vehicles (USVs) is an important issue for autonomous navigation of USVs. In this paper, an improved particle swarm optimization (PSO) is proposed for real-time autonomous navigation of a USV in real maritime environment. To overcome the conventional PSO’s inherent shortcomings, such as easy occurrence of premature convergence and human experience-determined parameters, and to enhance the precision and algorithm robustness of the solution, this work proposes three optimization strategies: linearly descending inertia weight, adaptively controlled acceleration coefficients, and random grouping inversion. Their respective or combinational effects on the effectiveness of path planning are investigated by Monte Carlo simulations for five TSPLIB instances and application tests for the navigation of a self-developed unmanned surface vehicle on the basis of multi-sensor data. Comparative results show that the adaptively controlled acceleration coefficients play a substantial role in reducing the path length and the linearly descending inertia weight help improve the algorithm robustness. Meanwhile, the random grouping inversion optimizes the capacity of local search and maintains the population diversity by stochastically dividing the single swarm into several subgroups. Moreover, the PSO combined with all three strategies shows the best performance with the shortest trajectory and the superior robustness, although retaining solution precision and avoiding being trapped in local optima require more time consumption. The experimental results of our USV demonstrate the effectiveness and efficiency of the proposed method for real-time navigation based on multi-sensor fusion.

## 1. Introduction

It is well known that multi-sensor fusion is an important issue for autonomous navigation of unmanned vehicles, especially when operating in real environments with unanticipated changes. With the aid of various types of sensors, such as temperature and humidity sensors, collision sensors, flow velocity and flow rate sensors, and displacement sensor, unmanned vehicles have been effectively applied to the fields of sounding survey [1], environment monitoring [2], underwater acoustics [3], marine rescue [4], target tracking [5], and water monitoring [6]. In these cases, all of the sensory information from multiple sensors is combined and effectively utilized to generate desirable trajectories for unmanned vehicles to follow, which is always formulated as a travelling salesman problem (TSP).

The TSP was proved to be a typical non-deterministically polynomially hard combination optimization problem in 1979 [7]. Its goal is to design the shortest route for a traveler to visit each city without repetition and ultimately return to the departure city. With the search-space tending to infinity and complexity, traditional exact algorithms, such as the enumeration method, fail to approach an exact solution within a reasonable computation time. Hence, novel algorithms with the capability of self-organization and self-adaption need to be developed to discover an adequate solution, sacrificing optimality, accuracy, and completeness for running speed. Inspired by natural evolution models and adaptive population evolution, collective intelligence methods, including genetic algorithm [8], particle swam optimization (PSO) [9], ant colony optimization (ACO) [10], artificial fish swarm algorithm [11], and artificial bee colony algorithm [12], have entered into a stage of rapid development for the TSP.

Particle swarm optimization, proposed by Eberhart and Kennedy in 1995, is an evolutionary metaheuristic technique [13]. It solves the optimization problem by having a population of candidate solutions, called particles, and moving these particles around in multi-dimensional search-space with a certain velocity. With a fitness function to assess each solution, the movements of all of the particles are dynamically guided by their own experience, as well as the entire swarm’s experience. Finally, it is expected that the swarm will move toward the most satisfactory solution. Due to advantages of fast convergence speed, simple parameter settings, and easy implementation, the PSO algorithm has been widely used in various fields, including functions optimization [14], training of neutral networks [15], and fuzzy system control [16]. 

Additionally, in order to improve the performance of PSO in solving the discrete-space-based TSP, valuable research has been conducted in recent times on the hybridization of heuristic methods. B. Shuang et al. proposed a hybrid algorithm that combined the respective advantages of PSO and ACO. The search mechanism of PSO was effectively utilized in which the particle’s experience helped to expand the search space, while the swarm experience pushed the global convergence [17]. X. Zhang et al. improved PSO by using a priority coding method to code the solution, dynamically setting the velocity range to remove the side effect due to the discrete search-space, and introducing the k-centers method to avoid the local optimum. The improved algorithm performed well in reserving the swarm diversity [18]. A hybrid fuzzy learning algorithm was proposed by H. M. Feng et al. in a large-scale search-space. The adaptive fuzzy C-mean algorithm was first used to divide the large-scale cities into subsets, following by the transform-based particle swarm optimization and the simulated annealing method acquiring the local optimal solution. Then the complete optimal route was rebuilt by the powerful MAX-MIN merging algorithm [19]. In the work by M. Mahi et al., the authors introduced the PSO into the ACO to help to optimize the city selection parameters, and the 3-opt algorithm was used for the purpose of jumping out of the local optimum [20]. In addition, the combination of PSO with genetic algorithm by W. Deng et al. [21], and the method combining PSO with artificial fish swarm algorithm [22] also show admirable improvement. 

It is well known that the PSO performance depends heavily on the proper balance between exploration, namely searching a broader space, and exploitation, namely, moving to the local optimum. It contends that tuning the PSO parameters has a significant impact on the optimization performance. Hence, choosing proper parameters to improve the algorithm effectiveness has been a hot spot for many works. In the work by Y. Zhang et al., the raw fitness value was adjusted by the power-rank scaling method, the acceleration coefficients and the inertia weight were changed with iteration, and the random numbers were modified to be generated by a chaotic operator. Simulation results showed that the novel method succeeded elite genetic algorithms with migration, simulated algorithm, chaotic artificial bee colony, and PSO in both success rate and time cost [23]. To solve the vehicle routing problem with time windows, a variant of PSO with three adaptive strategies was used, in which all parameters started with random values, but gradually tended to be applicable during iterations based on some limitations [24]. K. R. Harrison et al. analyzed the results of PSO using 3036 configurations of control parameters for 22 benchmark problems and found the time-dependence of optimal values. Meanwhile, the optimal range of acceleration coefficients and inertia weight were recommended [25].

Moreover, with the fast development of intelligent algorithms and autonomous navigation technology, PSO has also been successfully applied to the vehicle path planning problem. R. J. Kenefic combined a heading constraint heuristic with PSO to solve the turn rate limited TSP for an unmanned aerial vehicle. Permutations of the tour vertices’ orders were considered to eliminate the self-crossing phenomenon in the planned path. Results revealed that PSO performed better than a standard algorithm in MATLAB because of the discontinuous and multimodal nature of the objective function [26]. To plan the shortest and smoothest route for the robot, a novel algorithm was presented with the PSO component used as a global planner and the modified probabilistic road map method used as a local planner. Results showed that this PSO-based algorithm was advantageous in runtime and path length [27]. M. D. Phung et al. improved the PSO by integrating deterministic initialization, random mutation, and edge exchange. Experimental tests with real-world datasets from unmanned aerial vehicle inspection showed the proposed algorithm could enhance the performance in both computing time and travelling cost [28]. To plan a multi-objective optimization path for an autonomous underwater vehicle in dynamic environments, the PSO was used to find suitable temporary waypoints, combined with the waypoint guidance to generate an optimal path [29]. 

In order to avoid premature convergence, route self-crossing, and to enhance the robustness, this work proposes three improved algorithms on the basis of the PSO method by combining one or two optimization strategies as follows: linearly descending inertia weight, adaptively controlled acceleration coefficients, and random grouping inversion. First, a hundred Monte Carlo simulations are conducted for five TSPLIB instances in order to compare the effectiveness of each improved algorithm in terms of route length, computing efficiency and algorithm robustness. Furthermore, improved PSO algorithms are applied to the navigation, guidance and control system (NGC) of a self-developed USV with multi-sensor data in a real sea environment.

The main contributions of this work are as follows: (1) The important parameters, including the acceleration coefficients and inertia weight, are adjusted iteratively, with the aim of effectively reducing the path length and enhancing the robustness; (2) The strategy of random grouping inversion maintains the swarm diversity and accelerates the global convergence, which can avoid premature convergence and retain solution precision; (3) Path planning for a USV is conducted by combining the conventional PSO with the three optimization strategies, which generates feasible routes with satisfactory length and no self-crossing.

The rest of the paper is structured as follows. PSO algorithms with different optimization strategies are introduced concisely in Section 2. Results and discussions of Monte Carlo simulations and applications to a USV are presented in Section 3. Additionally, conclusions and future research directions are drawn in Section 4.

## 2. Proposed Algorithms

### 2.1. Particle Swarm Optimization

As mentioned in Section 1, the conventional PSO is a population-based stochastic optimization method. At the beginning of the evolutionary process, the PSO method generates *N* candidate solutions (namely *N* particles) randomly within an *S*-dimensional search space. For the *i*-th particle, its position can be represented by a vector *X_i_* = (*x_i_*_1_, *x_i_*_2_, ..., *x_iS_*)*^T^*. Meanwhile, its velocity can be defined by a vector *V_i_* = (*v_i_*_1_, *v_i_*_2_, ..., *v_iS_*)*^T^*. A fitness function is used to evaluate the quality of each solution. For the TSP and path planning problem in this work, the fitness function is defined as 1/*D* (*D* stands for the route length). For every iteration, all the particles depend on two kinds of experience for guiding their movement: the best position (*P_is_*) an individual has known so far, and the best position (*P_gs_*) the entire swarm has known so far. Correspondingly, the velocity and position of each particle are updated following Equations (1) and (2) [30].
(1)$$v_{is}^{m+1}=wv_{is}^{m}+c_{1}r_{1}^{m}\left(P_{is}^{m}-x_{is}^{m}\right)+c_{2}r_{2}^{m}\left(P_{gs}^{m}-x_{is}^{m}\right),$$
(2)$$x_{is}^{m+1}=x_{is}^{m}+v_{is}^{m+1},$$
where *m* and *s* stand for the current number of iterations and the *s*-th dimension, respectively. *r*_1_ and *r*_2_ are random and iteratively updated numbers uniformly distributed between 0 and 1. *c*_1_, *c*_2_, and *w* are PSO control parameters called personal cognition coefficient, social cognition coefficient, and inertia weight, respectively.

It should be noted that there are three terms of velocity on the right side of Equation (1). The first term $wv_{is}^{m}$ is the inertia component, which makes the particle move in its original direction of last iteration. The inertia weight *w*, first proposed by Y. Shi and R. C. Eberhart in 1998, affects the capability of global search and algorithm convergence, and it is typically set between 0.8 and 1.2 [30]. The second item $c_{1}r_{1}^{m}\left(P_{is}^{m}-x_{is}^{m}\right)$ is called the personal cognition component, which causes the particle to move according to its memory of individual best-known position. Meanwhile, the third item $c_{2}r_{2}^{m}\left(P_{gs}^{m}-x_{is}^{m}\right)$ is the social cognition component, which will guide the particle to move towards the swarm’s best known position based on communication with other particles. The acceleration coefficients *c*_1_ and *c*_2_ play an important role in balancing the effects of personal cognition and social cognition on guiding the particle towards the target optimal solution. The values of *c*_1_ and *c*_2_ are usually suggested to be 2. In addition, it is reported that the stochastic characteristics of *r*_1_ and *r*_2_ can weaken the effects of the individual best known position and the swarm best known position on the velocity update. The diversity of population could be maintained, and the phenomenon of premature convergence could be avoided to some degree [31].

Figure 1 shows a schematic diagram of position change of a particle for two successive iterations. The algorithm procedure will be terminated when the maximum number of iterations (*M*) or a minimum error threshold is achieved. The pseudo code of conventional PSO is presented in Algorithm 1. 

**Algorithm 1.** Conventional Particle Swarm Optimization for TSPselect swarm size and maximum iterationsdefine fitness functionpreset acceleration coefficients (*c_1_*, *c_2_*) and inertia weight (*w*)**for** each particle **do**    initialize velocity and position    evaluate initial fitness value    record initial *P_is_* and *P_gs_*
**end**
**while** maximum iterations or minimum error criteria is not achieved **do**    **for** each particle **do**       calculate the new velocity using Equation (1)       update the new position using Equation (2)       evaluate new fitness function       update *P_is_* and *P_gs_*    **end**
**end**


### 2.2. Linearly Descending Inertia Weight

Note that the inertia weight *w* reflects the effect of historical velocity on current velocity for each particle. It could balance the capacities of local and global searching. When *w* = 0, it could be found based on Equation (1) that the particle velocity only depends on its current cognition of the personal best-known position (*P_is_*) and the swarm’s best known position (*P_gs_*). If one particle is in its current *P_gs_*, it will remain stationary, while others will fly at a weighted speed of *P_is_* and *P_gs_*. Given this circumstance, the entire swarm will be pulled towards the current *P_gs_* and converge to the local optimum. On the contrary, with the aid of the inertia component, all the particles have a tendency to explore a larger space. Hence, when faced with various optimization problems such as functions optimization, training of neutral network, and Fuzzy system control, it is necessary to adjust the value of *w* to balance the algorithm capability of local and global searching.

In addition, the inertia weight *w* also affects the global search behavior, especially the convergence behavior. Generally, a lower value of *w* would help speed up the convergence of global optimum, while a larger value of *w* would contribute to the exploration of the entire search space. To obtain a better global search capability during early iterations, and enhance the local exploitation during later iterations to avoid being trapped into local optimum, the inertia weight *w* is adjusted dynamically with the form of linearly descending over the iterations according to Equation (3).
(3)$$w=w_{max}-m\times \left(w_{max}-w_{min}\right)/M$$
where *w_max_* and *w_min_* represent the maximum and minimum value of inertia weight *w*, respectively.

### 2.3. Adaptively Controlled Acceleration Coefficients

The acceleration coefficients *c*_1_ and *c*_2_ reflect the information exchange among particles, and determine the distance a particle will move towards target solution under the guidance of personal cognition and social cognition in a single iteration. Small values of acceleration coefficients would make the particle wander far from the target region, while large values of acceleration coefficients would urge the particle to move quickly towards the target region but ultimately deviate from this region. When *c*_1_ and *c*_2_ are both equal to zero, the particle will fly at its current velocity until it hits the border of the search space. As a result, the satisfactory solution is hard to find within the restricted search space. If *c*_1_ is zero, the particle will lose cognitive function. Although the search space could be enlarged by taking into account the particle interactions, it is more likely to be trapped in the local optimum when faced with a complex optimization problem. In addition, when *c*_2_ is zero, no information exchanges exist in the swarm; each particle will work independently. It is almost impossible to find the optimal solution.

As mentioned in Section 2.1, the values of *c*_1_ and *c*_2_ keep constant during the whole evolutionary procedure for the conventional PSO. However, the fixed settings have inherent limitations: large values make each particle rapidly converge towards the local optimum, while low values cause each particle to wander far from target regions. Hence, a concept of iteratively linearly changing acceleration coefficients was employed by A. Ratnaweera et al. [32], Y. Zhang et al. [23], and Z. Yan et al. [29]. A relatively larger *c*_1_ and a relatively lower *c*_2_ were used during the early stage of iterations. With the increasing iterations, the value of *c*_1_ was linearly reduced, while the value of *c*_2_ was linearly increased, as formulated by Equations (4) and (5). It was reported that the linear-changing acceleration coefficients could help to reduce the probability of premature convergence during early iterations, and also enhance the convergence performance during later iterations.
(4)$$c_{1}=\left(c_{1min}-c_{1max}\right)\frac{m}{M}+c_{1max},$$
(5)$$c_{2}=\left(c_{2max}-c_{2min}\right)\frac{m}{M}+c_{2min},$$
where the subscripts *max* and *min* stand for the maximum and minimum values of acceleration coefficients *c*_1_ and *c*_2_.

However, it is thought that the effects of acceleration coefficients on algorithm convergence are restricted when their values are changing linearly during the whole evolutionary procedure. For instance, a larger influence of social information is of great significance in later stage of algorithm to improve the searching efficiency, which could not be realized timely by the simple linear variations of acceleration coefficients. Consequently, an evaluation parameter (*K*) is introduced in this work according to the degree of swarm convergence. Its value is defined as the radio of the number of successfully converged particles (called advantageous particles) in a single iteration to the initial swarm size, as represented in Equation (6). Then the evaluation parameter (*K*) is employed to adaptively control the changing rate of acceleration coefficients by using Equations (7) and (8).
(6)K=P/N,
(7)$$c_{1}=\left(c_{1max}-c_{1min}\right)K+c_{1min},$$
(8)$$c_{2}=\left(c_{2max}-c_{2min}\right)K+c_{2min},$$
where *P* stands for the number of particles that could successfully converge in a single iteration. The strategy of adaptively controlled acceleration coefficients associates the values of acceleration coefficients with the optimization status by use of the evaluation parameter (*K*). With the increase of iterations, the number of advantageous particles in a swarm is increasing; this will enlarge the influence of advantageous particles on the entire swarm. Hence, it is supposed that excellent solutions would be protected as much as possible to help avoid local optima.

### 2.4. Random Grouping Inversion

It should be noted that the CPSO uses a single swarm consisting of all the particles for evolution. Hence, it is likely to result in a phenomenon in which all the particles cluster around a certain position and stop exploring the other area of the search-space. To avoid the easy occurrence of premature convergence, the concept of random grouping inversion is proposed and added before the update of *P_is_* and *P_gs_* during every iteration. The single swarm is divided into several subgroups, in which independent evolution is in process. As a result, the diversity of the swarm can be strengthened and the global convergence for the entire swarm is accelerated. 

As to the number of particles in a subgroup, preliminary research indicated that a larger number would decrease the inherent capability of merit-based selection, while a lower number would weaken the role of grouping mechanism. Finally, the number was set as four; in other words, four particles were randomly sorted to form a subgroup.

On the basis of the random grouping strategy, a further operation is proposed, simulating the inversion operation during the process of biologic evolution. After evaluating the fitness of each particle, all four particles clustered around the local optimum discovered by each subgroup. Then the inversion was carried out to generate new particles and replace two original particles of the subgroup, in which the TSP tour orders for two randomly selected inversion points were inversed. 

Indeed, the strategy of random grouping inversion is based on Darwin’s theory of evolution: internal competition of population and uncertain mutation. In theory, the internal competition of the population is a process of merit-based selection, namely, only the fittest one survives. The inversion is a type of uncertain mutation which could help to maintain the swarm diversity. Ultimately, the pool of swarm particles reserves not only the fittest individual of each subgroup, but also the inversion-based variant. Hence, it is supposed that this strategy would help to enhance the population diversity and improve the effectiveness of swarm optimization. The pseudo code of random grouping inversion is shown in Algorithm 2.

**Algorithm 2.** Random Grouping InverionRandperm swarm size**for** each subgroup    find the fittest particle among four particles    randomly select inversion points    inversion    update *P_gs_*
**end**


## 3. Simulations and Experimental Results

### 3.1. Monte Carlo Simulations

Three algorithms with progressively improved strategies are proposed based on the conventional particle swarm optimization (CPSO): the algorithm with adaptively controlled acceleration coefficients (APSO), the algorithm with both adaptively controlled acceleration coefficients and linearly descending inertia weight (AWPSO), and the algorithm combining the advantages of adaptively controlled acceleration coefficients, linearly descending inertia weight, and random grouping inversion (AWIPSO). To eliminate the stochastic property of PSO in MATLAB operation environment, this section employs Monte Carlo simulations to compare the performance of each algorithm for TSP from three points of view: the number of planned points, the swarm size, and the computing efficiency. All the simulations are performed on the same computer (Intel (R) Core (TM) i7-7700HQ CPU @ 2.80 GHz) with 8.0 GB memory so as to avoid the effects of computer models on the running capacity of algorithms. In addition, all the sample instances are taken from TSPLIB (Website: https://wwwproxy.iwr.uni-heidelberg.de/groups/comopt/software/TSPLIB95/tsp/#opennewwindow).

#### 3.1.1. Comparative Study with Different Numbers of Planned Points

Five considered instances from TSPLIB are burma14, ulysses22, eil51, eil76, and rat99. Correspondingly, the numbers of planned points (*Q*) are 14, 22, 51, 76, and 99, and the maximum numbers of iterations (*M*) are set as 100, 200, 1600, 2000, and 2000, respectively. The swarm size (*R*) is set as 500 for each algorithm. For the CPSO, acceleration coefficients *c*_1_ and *c*_2_ are constant at a value of 2, and *w* is also unchanged with the value of 0.9. For the improved algorithms, the varying range of the personal cognition coefficient *c*_1_ is set as 0.9–1.2, and the social cognition coefficient *c*_2_ varies from 0.2 to 1.0 during evolutionary procedure. As to the linearly descending inertia weight *w*, the value is changing from 0.9 to 0.4. Detailed information of parameter settings for each algorithm is listed in Table 1.

Monte Carlo simulations are conducted one hundred times to obtain the data set of optimal path distance (*D*) with four algorithms for each TSP instance. The comparative results are presented in box-and-whisker plots (see Figure 2). The legend and explanation of the box-and-whisker plots refer to the work by M. E. Spear [33]. In every box plot, a range bar represents the interquartile range of the data set, which indicates the degree of data dispersion and the algorithm robustness to some extent. The median value and the average value are identified with a red line and a plus symbol in the bar. Meanwhile, the whiskers are drawn extending on both sides of the bar, with the ends standing for the best and worst values, respectively. Furthermore, Table 2 lists the detailed statistics of the data set for each algorithm and each number of planned points, including the known optimal solution of TSPLIB, the worst, the best and the average values of optimal path distance. Additionally, the standard deviation is calculated to show how far the set of data is spread out from the average value, and it stands for the robustness of the algorithm.

In the instance of burma14 in Figure 2a, it is observed that the proposed optimization strategies help reduce the average values of *D*, and make the data set cluster more closely. However, a minor difference less than 0.4% is found in terms of path distance and algorithm robustness for the three improved algorithms. 

When the number of planned points (*Q*) increases (see Figure 2b–e), the respective advantages of the adaptively controlled acceleration coefficients, the linearly descending inertia weight, and the random grouping inversion emerge gradually. For *Q* = 76 in Figure 2d, the strategy of adaptively controlled acceleration coefficients plays a substantial role in reducing the path length by approximate 17.7% when comparing the APSO with CPSO. Meanwhile, the linearly descending inertial weight mainly affects the degree of data dispersion. The AWPSO has a standard deviation of 52.48 m, which is 10.7% lower than the APSO. By contrast, the AWIPSO is superior, with the shortest average path distance of 590.1 m and the lowest standard deviation of 12.4 m. It can be concluded that the strategy of random grouping inversion has considerable effects on both reducing the path length and improving the algorithm robustness. In addition, the best value of AWIPSO is 1248 m for *Q* = 99, which is only 3% larger than the known optimal solution of TSPLIB. 

#### 3.1.2. Comparative Study with Different Swarm Sizes

In this section, the TSP instance of eil51 with 51 planned points is selected as the working condition. Five swarm sizes (*R*) of 300, 400, 500, 600, and 700 are considered. Furthermore, the maximum number of iterations (*M*) for each algorithm is set as 500. The other parameter settings of each algorithm, such as *c*_1_, *c*_2_, and *w*, are the same as those in Section 3.1.1. 

The Monte Carlo simulations of one hundred times are conducted for each algorithm and each swarm size. Comparative results are shown in the form of five box-and-whisker plots in Figure 3. Detailed statistics of optimal path distance, including the four-number summary of data sets, are listed in Table 3.

When the swarm size is 300, as shown in Figure 3a, the respective advantages of the three optimization strategies could be concluded similarly as in Figure 2. The adaptively controlled acceleration coefficients contribute to shortening the path distance. The linearly descending inertia weight has a certain effect on reducing the degree of data dispersion by 2.1% when comparing the AWPSO with APSO. The AWIPSO performs best with an average optimal path length of 457.0 m and a standard deviation of 9.9 m. Compared with the AWPSO, the effects of random grouping inversion are clear in both shortening the path distance and improving the algorithm robustness. In addition, the median value is almost coincided with the mean value in each bar; this means all the algorithms could produce uniformly distributed data for the condition of eil51.

In general, the effects of the swarm size on algorithm performance are that the overall optimal distance can be further reduced for each algorithm with the increase of swarm size. For instance, the average value of D for the CPSO is 785.0 m for *R* = 700, which is 10% reduced compared with the case of *R* = 300. For the AWIPSO, the average D is 456.1 m when *R* = 700, and is only reduced by 0.2% with respect to the case of *R* = 300. It could be concluded that the effects of swarm size on algorithm performance are not evident for the AWIPSO. Furthermore, although the robustness of every algorithm changes a little due to the swarm size, no regular tendency could be found. By contrast, the AWIPSO is always the most advantageous algorithm. Both the optimal path length and the algorithm robustness are almost unaffected by the swarm size. In the case of *R* = 700, the mean distance and the standard deviation of AWIPSO are 456.1 m and 9.2 m, which are 42.0% and 78.3% smaller than that of CPSO, respectively.

#### 3.1.3. Comparative Results of Computing Efficiency

This section presents the evolution curves of five TSPLIB instances with different planned points (see in Section 3.1.1) to compare the computing efficiency of each algorithm. Two main criteria are selected for efficiency evaluation: the time consumption and the convergence speed. The former means the time consumption of completing the maximum number of iterations, and the latter refers to the critical number of iterations (*m**cri*) at which the solution converges to the best value. Figure 4 shows the evolution history of optimal path distance (*D*) against iteration (*m*) for each algorithm within five hundred iterations. Meanwhile, all of the detailed information regarding computing efficiency for each algorithm is listed in Table 4.

Generally, it can be observed that the evolution curve of optimal path distance for each algorithm declines sharply with the increase of iteration during the initial stage, then the declining trend becomes a little milder, and finally reaches to a horizontal level at a critical number (*m**cri*). The development of the evolution curve and the value of global optimum distance are completely dependent on the number of planned points and the utilized algorithm. With the increase of planned point numbers (*Q*), both the critical number and the optimal distance have an increasing tendency for each algorithm.

By contrast, the AWIPSO converges to the shortest path distance when compared with the other algorithms for five numbers of planned points considered in this section; this was described in Section 3.1.1. In addition, when *P* is less than 22, the critical number for AWIPSO has no evident difference compared with the others. However, with the increase of *Q*, the largest number of iterations is needed by the AWIPSO for convergence. When *Q* = 76, for instance, the *m_cri_* of AWIPSO is 378, which is 6, 1.9 and 1.4 times larger than that of CPSO, APSO, AWPSO, respectively. The cause behind this may be that the condition with more planned points would increase the complexity of route, and has a stronger demand for the algorithm performance, especially in avoiding the premature convergence. To put it another way, the CPSO, the APSO and the AWPSO are likely to be trapped in the local optimum during the evolution process which results in a relatively smaller *m_cri_*, while the AWIPSO could maintain the precision of solution which needs more iterations before convergence. As to the computing time, it is evident that the AWIPSO spends 6.7, 2.4 and 2.4 times more time than the CPSO, the APSO and the AWPSO to complete the same number of iterations when *Q* = 99. To reduce the path distance, it is necessary to extend the computation time cost to a certain degree.

Furthermore, Figure 5 presents the best trajectories of the five TSPLIB instances (burma14, ulysses22, eil51, eil76, rat99) using the AWIPSO. The abscissa and ordinate stand for the values of latitude and longitude, respectively. The start point is enclosed in the red rectangle, and the arrows represent the heading of the planned path. It can be observed that the trajectories become more complex in the path shape and distance as *Q* increases.

### 3.2. Multi-Sensor-Based Application to Unmanned Surface Vehicle

Recently, USVs have been utilized worldwide in both civil and military fields, such as spot cruises in ocean ranching and multi-point water quality monitoring and sampling in vast water, due to the benefits of reduced casualty risk and increased mission efficiency [34,35,36]. A USV is typically equipped with a motion control unit, a sensor unit, a communications unit and an arming system. The motion control unit consists of the navigation positioning subsystem, the path planning subsystem and the trajectory tracking subsystem. As a core technology, the path planning is of great significance in realizing the autonomous navigation and control of the USV. 

Generally, path planning can be reduced to the TSP if the prior environmental information is inaccessible and the collision-free restriction is not taken into account. Hence, the aforementioned algorithms can also be used to solve the USV path planning problem. In this section, the effectiveness of the improved algorithms is proved again by the application to the NGC system of a self-developed USV in real sea environment. As a preliminary study, the present work neglects the factors of wind, current and waves in the algorithms.

#### 3.2.1. Unmanned Surface Vehicle Model and Multi-Sensors

Figure 6 presents a 3D model and physical photo of the USV model, which is self-designed and constructed by the Sea Wolf group of Qingdao University of Science and Technology. The length and width of the USV model are 1.8 m and 0.9 m, respectively. It has five side bodies submerged in the water. Meanwhile, the electrical motors require a 48 V 45 A battery to provide power for the propeller.

The NGC system, as shown in Figure 7, involves three module subsystems: the navigation data processing subsystem, the path planning subsystem, and the autopilot subsystem. In addition, multi-sensors such as electronic compass and GPS are employed for gathering the navigation information of the direction of the bow and the location data of the USV. A WeatherStation^®^ PB200 ultrasonic weather sensor, produced by AIRMAR^®^ Technology Corporation, is used to collect the real-time, site-specific weather and location information. The navigation data are acquired by a navigation data acquisition system in real-time, along with the ship’s log and the status information.

Then, all the information is transferred to the path planning subsystem, where the PSOs are used to generate a desirable trajectory for the USV to follow. The autopilot employs a closed-loop controller to determine the heading and speed of the USV. In addition, a graphical user interface program is compiled based on the Spring model view controller framework to process and record all the data in a personal computer. The GPRS wireless network is established as the communication unit with the effective distance of 5 km. The transmission speed is 1–100 Mbps [37]. 

#### 3.2.2. Application Tests in Real Maritime Environment

This section applies the four aforementioned PSO methods to the NGC system of a USV model in real sea environment near the Qingdao Olympic Sailing Center in Fushan Bay. Two working conditions are used with different numbers of planned points: *Q* = 35 and *Q* = 45. The condition of *Q* = 35 has the start point of (N 36°03′45.71″, E 120°25′57.18″) in latitude and longitude, and the condition of *Q* = 45 starts from the point of (N 36°03′45.78″, E 120°25′56.66″). The location coordinates of other planned points refer to Table A1 and Table A2. Then, comparative studies are conducted to assess the effectiveness of path planning for each algorithm. The swarm size (*R*) is set as 500. The maximum numbers of iterations (*M*) are 350 and 450, respectively, which are dependent on the numbers of planned points.

Figure 8 and Figure 9 present the optimal trajectory of each algorithm under each working condition with the detailed information listed in Table 5. When *Q* = 35 in Figure 8, it is clear that the CPSO provides the worst route with serious level of path-crossing phenomenon, as marked in black circles, when compared with the other algorithms. This would be the reason the longest route distance is generated by the CPSO under the same condition. By using the three optimization strategies, it is possible to remove the self-crossing in the path to different degrees. Moreover, the AWIPSO performs best, with the shortest path of 1229.88 m and a relatively lower *m_cri_* of 99. Using pair-wise comparison, it is discovered that the path length has been optimized by 21.4%, 14.7%, and 5.6% for strategies of adaptively controlled acceleration coefficients, linearly descending inertial weight, and random grouping inversion, respectively.

When *Q* = 45, it could be found that the trajectories generated by the CPSO, APSO, and AWPSO have different levels of self-crossing in Figure 9a–c; this results in the evident increase of path length. However, at the same time, the advantages of the AWIPSO reflect more obviously in avoiding the intersection of route effectively and simplifying the path shape, especially when more planned points are considered. The optimal path length generated by the AWIPSO is 1380.84 m, which is 48.0%, 38.9%, and 27.5% shorter than that of the CPSO, APSO, and AWPSO, respectively. Pair-wise comparison indicates that the optimization with three rates of 14.8%, 15.7%, and 27.5% were made by the respective effect of adaptively controlled acceleration coefficients, linearly descending inertial weight, and random grouping inversion. 

In addition, all the improved algorithms require more time for computation than the CPSO under the same maximum number of iterations. As mentioned in Section 3.1.3, it is necessary to extend the time-cost to a certain degree for the purposes of improving the precision of solution and avoiding being trapped into local optima.

## 4. Conclusions

Multi-sensor fusion is an important issue for autonomous navigation of USVs. This work proposes three optimization strategies based on the conventional particle swarm optimization for real-time autonomous navigation of a USV in real maritime environment: the linearly descending inertia weight, the adaptively controlled acceleration coefficients, and the random grouping inversion. Monte Carlo simulations for five TSPLIB instances and application tests to an unmanned surface vehicle are conducted to reveal their respective or combinational advantages. Results can be concluded as follows:(1)The adaptively controlled acceleration coefficients employ the influence of advantageous particles on the swarm, enhancing the capacity of the global search during early iterations and the local search during later stages. The strategy plays a substantial role in reducing the path length.(2)The linearly descending inertia weight mainly helps to improve the algorithm robustness.(3)The random grouping inversion optimizes the capacity of local search and maintains the population diversity; this can avoid premature convergence and keep the solution precision.(4)The PSO combined with all the three strategies is superior to generate routes with the most satisfactory length and no self-crossing. However, more time consumption is required before global convergence.(5)With more planned points, a more complex trajectory would be generated, would have a strong demand in terms of algorithm performance. However, the effects of swarm size on path planning for each algorithm are irregular, which could be neglected to some extent.

Since this work carries out the preliminary studies in optimizing the conventional PSO, more efforts are needed especially in further reducing the algorithm time cost. In the future, more comparative studies with different optimization algorithms for TSP will be conducted. 

## Figures and Tables

**Figure 1 sensors-19-03096-f001:**
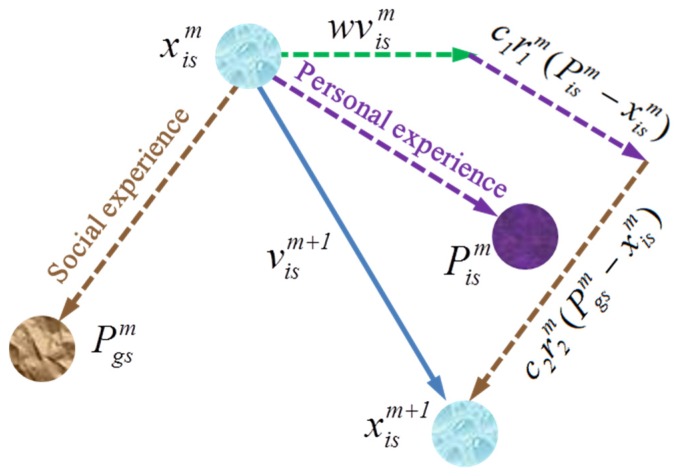
Shematic diagram of a particle’s position update in PSO.

**Figure 2 sensors-19-03096-f002:**
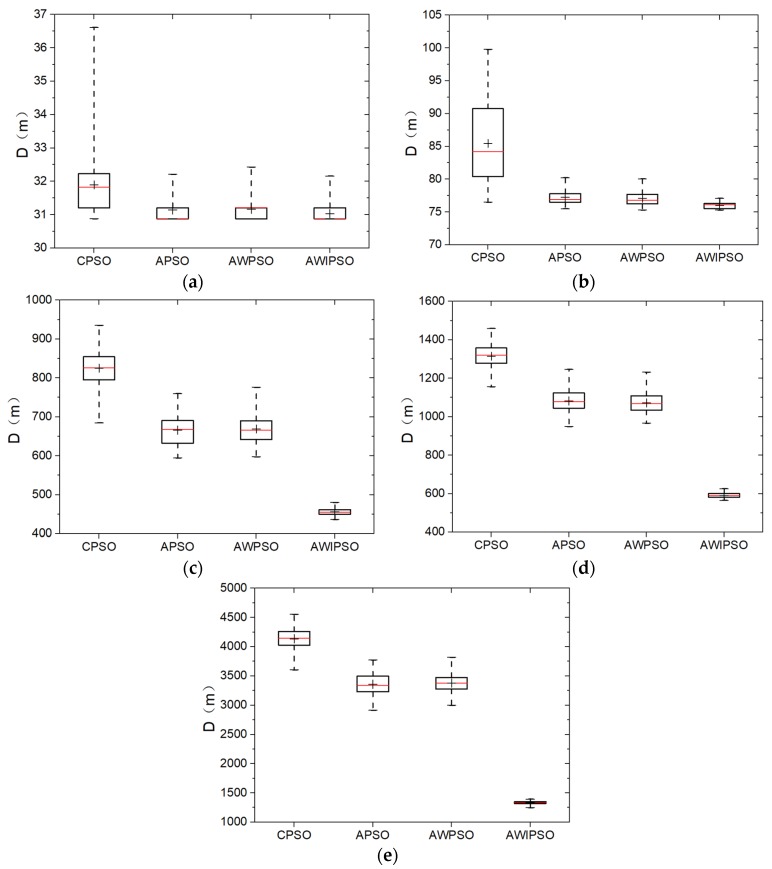
Solution distribution of each algorithm with five numbers of planned points: (**a**) *Q* = 14; (**b**) *Q* = 22; (**c**) *Q* = 51; (**d**) *Q* = 76; (**e**) *Q* = 99.

**Figure 3 sensors-19-03096-f003:**
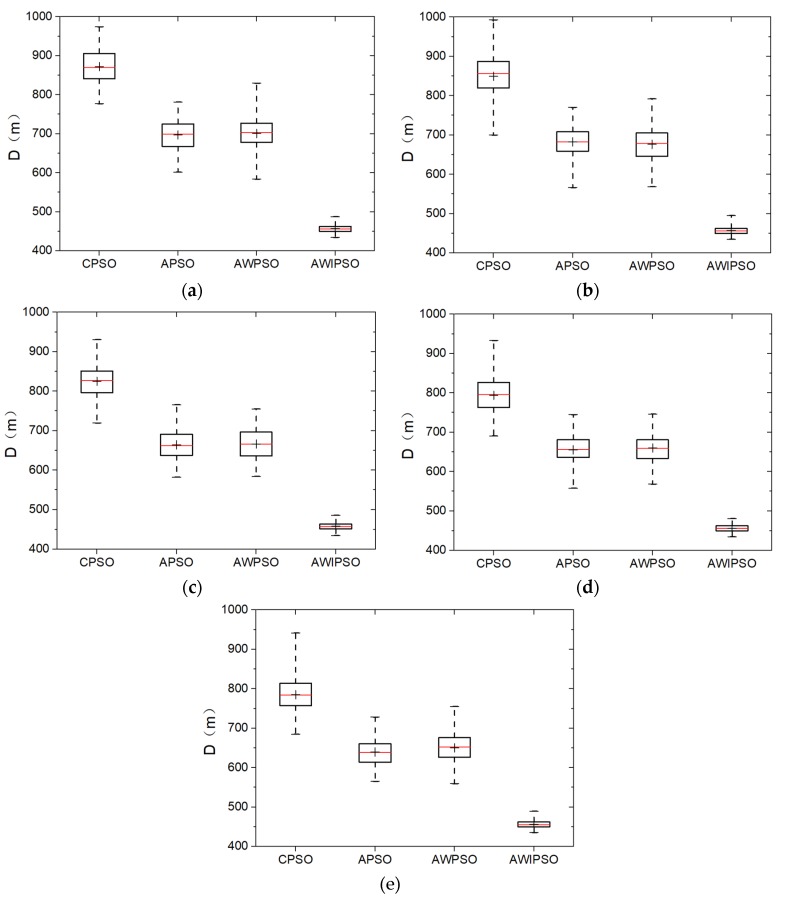
Solution distribution of each algorithm with five population sizes: (**a**) *R* = 300; (**b**) *R* =400; (**c**) *R* = 500; (**d**) *R* =600; (**e**) *R* =700.

**Figure 4 sensors-19-03096-f004:**
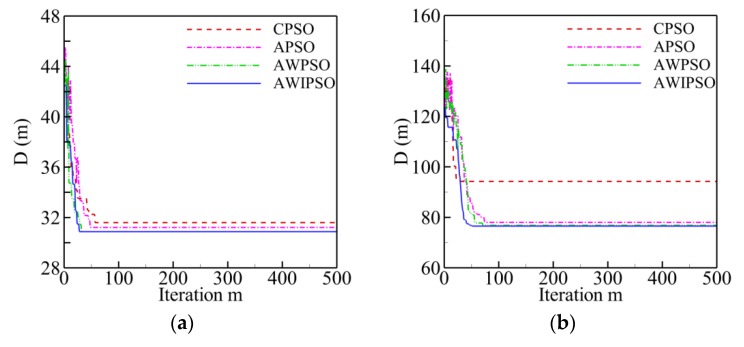
Evolution curves of optimal path distance against iterations for each algorithm: (**a**) *Q* = 14; (**b**) *Q* = 22; (**c**) *Q* = 51; (**d**) *Q* = 76; (**e**) *Q* = 99.

**Figure 5 sensors-19-03096-f005:**
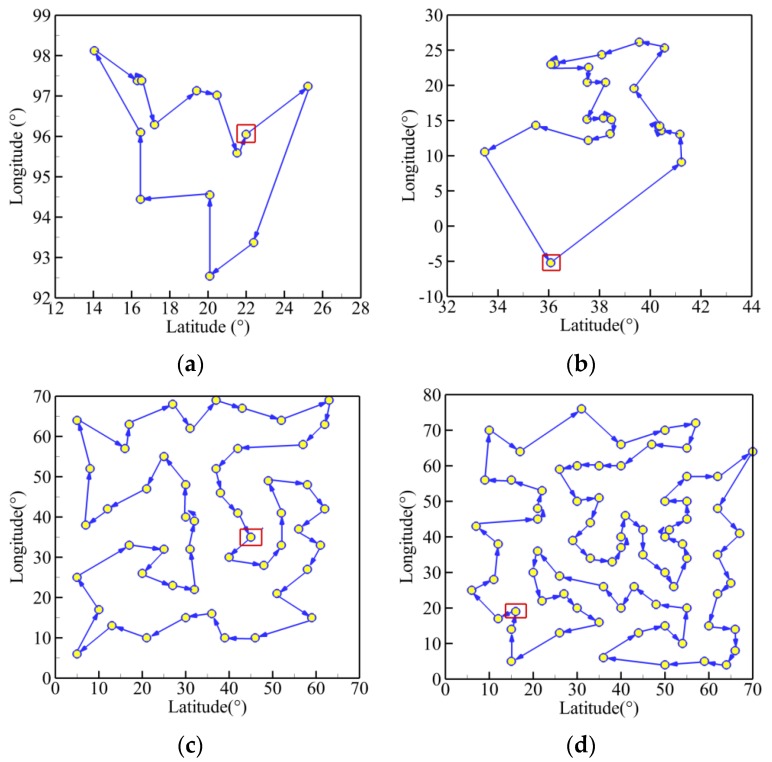
Best routes of five TSPLIB examples: (**a**) burma14; (**b**) ulysses22; (**c**) eil51; (**d**) eil76; (**e**) rat99.

**Figure 6 sensors-19-03096-f006:**
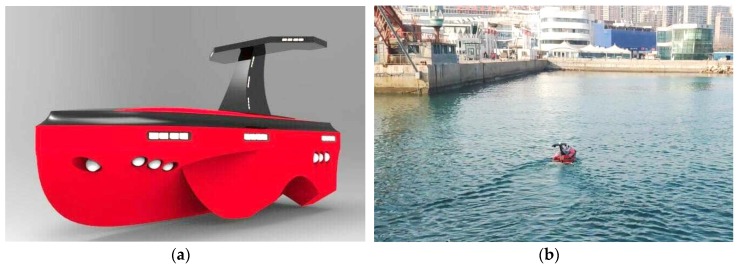
A self-developed unmanned surface vehicle: (**a**) 3D model; (**b**) USV in water.

**Figure 7 sensors-19-03096-f007:**
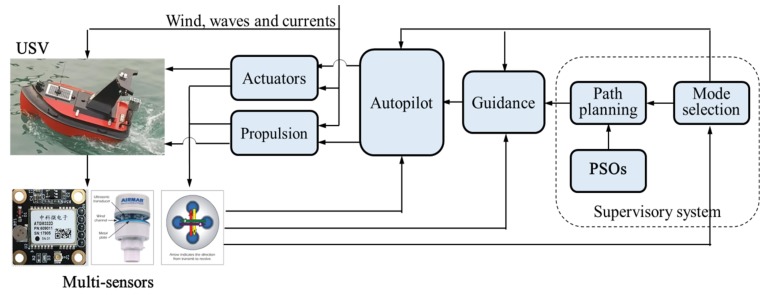
Navigation, guidance and control system of USV.

**Figure 8 sensors-19-03096-f008:**
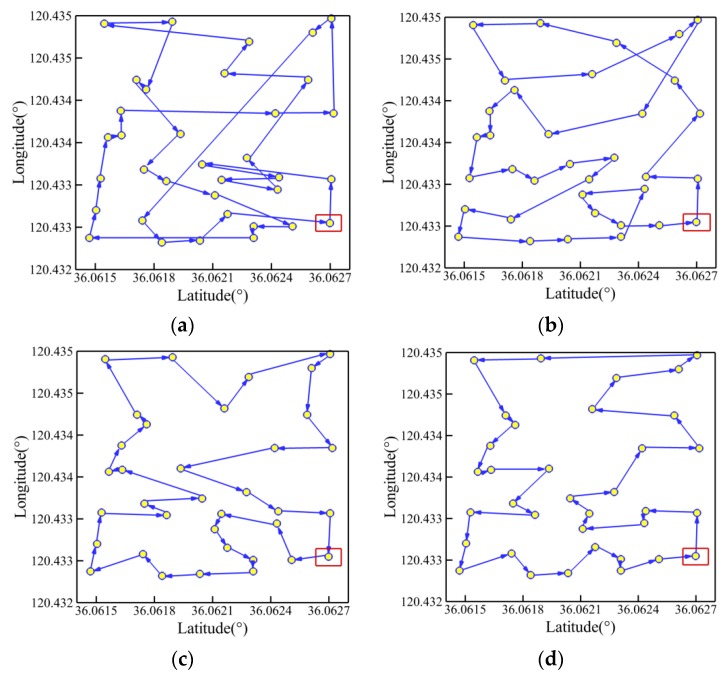
Trajectory planned by each algorithm for *Q* = 35: (**a**) CPSO; (**b**) APSO; (**c**) AWPSO; (**d**) AWIPSO.

**Figure 9 sensors-19-03096-f009:**
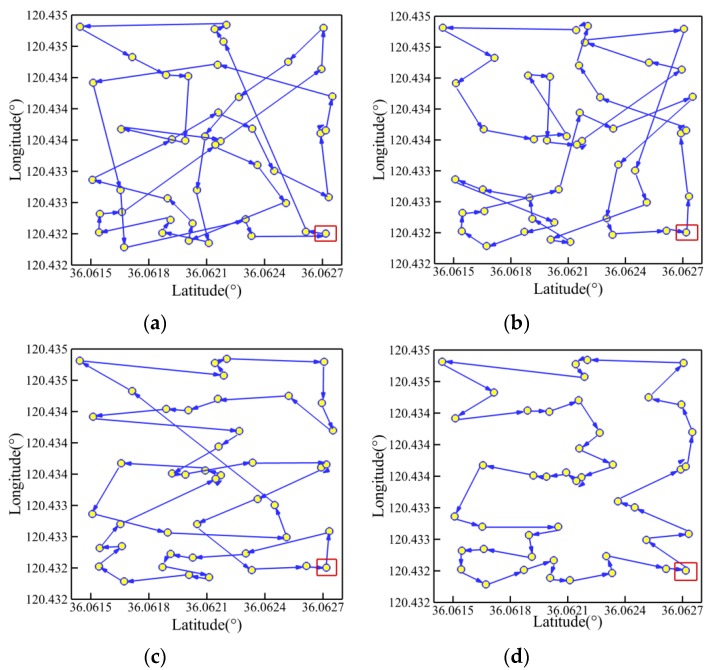
Trajectory planned by each algorithm for *Q* = 45: (**a**) CPSO; (**b**) APSO; (**c**) AWPSO; (**d**) AWIPSO.

**Table 1 sensors-19-03096-t001:** Parameter settings for each algorithm.

Algorithm	Parameter	Setting
CPSO	personal cognition coefficient *c*_1_	constant, 2
	social cognition coefficient *c*_2_	constant, 2
	inertia weight *w*	constant, 0.9
APSO	personal cognition coefficient *c*_1_	adaptively controlled by Equation (7), 0.9–1.2
	social cognition coefficient *c*_2_	adaptively controlled by Equation (8), 0.2–1.0
	inertia weight *w*	constant, 0.9
AWPSO	personal cognition coefficient *c*_1_	adaptively controlled by Equation (7), 0.9–1.2
	social cognition coefficient *c*_2_	adaptively controlled by Equation (8), 0.2–1.0
	inertia weight *w*	linearly descending by Equation (3), 0.9–0.4
AWIPSO	personal cognition coefficient *c*_1_	adaptively controlled by Equation (7), 0.9–1.2
	social cognition coefficient *c*_2_	adaptively controlled by Equation (8), 0.2–1.0
	inertia weight *w*	linearly descending by Equation (3), 0.9–0.4

**Table 2 sensors-19-03096-t002:** Statistical results of optimal path distance in 100 runs with five numbers of planned points.

*Q*	Known Optimal Solution	Algorithm	Worst (m)	Best (m)	Mean (m)	Std. Dev. (m)
14	30.88 m	CPSO	36.61	30.88	31.90	1.00
		APSO	32.21	30.88	31.14	0.36
		AWPSO	32.42	30.88	31.16	0.38
		AWIPSO	32.15	30.88	31.03	0.29
22	74 m	CPSO	99.78	76.49	85.45	6.22
		APSO	80.22	75.50	77.26	1.13
		AWPSO	80.05	75.30	77.11	1.10
		AWIPSO	77.10	75.30	75.97	0.46
51	426 m	CPSO	935.06	684.73	825.02	45.67
		APSO	759.91	594.27	665.87	38.96
		AWPSO	775.86	597.08	668.68	38.72
		AWIPSO	480.14	436.06	455.91	9.37
76	538 m	CPSO	1459.78	1156.23	1315.03	62.90
		APSO	1247.22	948.52	1082.12	58.75
		AWPSO	1231.88	966.33	1071.70	52.48
		AWIPSO	625.39	564.86	590.11	12.40
99	1211 m	CPSO	4555.13	3602.25	4136.63	194.64
		APSO	3772.27	2912.69	3359.80	178.09
		AWPSO	3819.77	2998.28	3379.08	173.14
		AWIPSO	1394.07	1248.12	1332.97	27.62

Std. Dev. is the abbreviation for standard deviation.

**Table 3 sensors-19-03096-t003:** Statistical results of optimal path distance in 100 runs with five swarm sizes.

*R*	Algorithm	Worst (m)	Best (m)	Mean (m)	Std. Dev. (m)
300	CPSO	974.27	776.77	872.24	43.50
	APSO	781.48	601.64	696.70	40.75
	AWPSO	829.51	583.85	700.67	39.90
	AWIPSO	487.43	434.33	457.03	9.85
400	CPSO	992.82	699.91	849.28	51.74
	APSO	770.40	566.11	682.41	37.41
	AWPSO	792.31	568.42	676.16	41.31
	AWIPSO	495.28	434.84	456.84	10.69
500	CPSO	930.74	719.46	825.04	40.90
	APSO	765.91	582.15	663.66	35.23
	AWPSO	755.14	584.22	666.19	37.57
	AWIPSO	485.47	434.60	457.57	9.75
600	CPSO	933.20	690.43	793.78	49.21
	APSO	744.87	557.78	655.20	32.53
	AWPSO	746.03	568.09	659.74	38.71
	AWIPSO	480.44	434.9	456.19	9.11
700	CPSO	941.57	684.87	784.95	42.51
	APSO	728.26	564.87	639.80	34.03
	AWPSO	755.14	559.05	650.16	37.26
	AWIPSO	489.62	435.14	456.05	9.23

Std. Dev. is the abbreviation for standard deviation.

**Table 4 sensors-19-03096-t004:** Simulating results of computing efficiency for each algorithm.

*Q*	*M*	Algorithm	*m_cri_*	Time Cost
14	100	CPSO	57	0.3
		APSO	48	0.4
		AWPSO	32	0.3
		AWIPSO	28	1.3
22	200	CPSO	32	0.4
		APSO	70	0.8
		AWPSO	75	0.9
		AWIPSO	51	2.9
51	1600	CPSO	57	1.5
		APSO	153	11.0
		AWPSO	186	4.2
		AWIPSO	264	13.7
76	2000	CPSO	61	3.0
		APSO	202	8.3
		AWPSO	266	10.6
		AWIPSO	378	23.6
99	2000	CPSO	67	5.0
		APSO	210	14.0
		AWPSO	222	14.1
		AWIPSO	275	33.7

**Table 5 sensors-19-03096-t005:** Simulation results of each algorithm with two numbers of planned points.

*Q*	*M*	Algorithm	*m_cri_*	Time Cost (s)	*D* (m)
35	350	CPSO	56	1.66	1942.50
		APSO	147	2.97	1527.36
		AWPSO	165	3.41	1303.14
		AWIPSO	99	6.55	1229.88
45	450	CPSO	60	2.45	2654.01
		APSO	155	4.11	2261.07
		AWPSO	217	5.17	1904.76
		AWIPSO	118	10.64	1380.84

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
