# Peer review of "Application of Improved Particle Swarm Optimization for Navigation of Unmanned Surface Vehicles"

_sensors, 2019, doi:10.3390/s19143096_

Round 1
Reviewer 1 Report
The paper is nicely written and has been improved in the revision.
I found some English errors in the current version. The authors are encouraged to revise the English to improve the writing.
Some examples:
Page 2: all the sensory information from multiple sensors are (is) combined
Page 3: valuable research have (has) been conducted
Page 3: In the work by M. Mahi et al. (the authors) introduced the PSO
Page 4: algorithm effectiveness has been a hot spot for many research (works/papers)
Page 9: with the particle interaction taken into account--> by taking into account the particle interactions
Page 23: The navigation data is (are) acquired
Page 26: in the path to different degree (degrees)
Author Response
We really appreciate all insightful comments and useful suggestions, which definitely will help us to improve the quality and readability of our manuscript. The manuscript has been revised according to the comments. In order to make the revising contents more clearly, the changes according to the reviewer#1 and reviewer#2 comments for the grammar, spelling and syntax problems will be highlighted in Blue; the changes according to the reviewer#3 comments will be highlighted in Yellow.
Reviewer #1
The paper is nicely written and has been improved in the revision. I found some English errors in the current version. The authors are encouraged to revise the English to improve the writing.
Some examples:
Page 2: all the sensory information from multiple sensors are (is) combined
Page 3: valuable research have (has) been conducted
Page 3: In the work by M. Mahi et al. (the authors) introduced the PSO
Page 4: algorithm effectiveness has been a hot spot for many research (works/papers)
Page 9: with the particle interaction taken into account--> by taking into account the particle interactions
Page 23: The navigation data is (are) acquired
Page 26: in the path to different degree (degrees)
Response: Thanks for the reviewer’s critical comment. Following the reviewer’s suggestion, we carefully checked the manuscript entirely ourselves to eliminate all the simple grammar errors such as the wrong usage of the verb conjugations and other sentences which are not easy to be understood. All the responses in this file are also carefully checked for smoothness. Please checked all the highlighted changings in the revised manuscript.
Reviewer 2 Report
The paper addresses the problem of autonomous navigation of Unmanned Surface Vehicles (USVs) by enhancing conventional Particle Swarm Optimization (PSO) techniques. The proposed enhancements include: (i) Linear descend of inertia weight, (ii) Adaptive control of acceleration coefficients, and (iii) Random grouping inversion. These enhancements are combined into three algorithms: APSO (ii), AWPSO (i)+(ii) and AWIPSO (i)+(ii)+(iii). These algorithms are compared against conventional PSO in three metrics: trajectory length, CPU time, and swarm size.
The paper is well written (although there are some small issues - see below) and the presented enhancements of the conventional PSO are easy to understand. The experiments show that AWIPSO generates best quality solutions although taking more CPU time. On top of that, the approaches are validated in a real-world scenario. I don't have much to criticize in general and hence I suggest "minor revisions".
Detailed comments:
- "reducing the optimal path length" (Introduction) - Optimal path length means the shortest path length, hence it cannot be more reduced. Deleting the word "optimal" from that sentence will do the job.
- Tables 5 and 6 give a list of coordinates that (I believe) are depicted in Figures 9 and 10. As the tables are hard to read for humans, they can be removed.
- There are missing references (page 10 top) of Zhang et al. and Yan et al.
- "non-deterministic polynomial hard" (page 2) -> "non-deterministicaly polynomially hard" or better "NP-hard"
- Missing space before "In order to plan..." (page 4 bottom)
- "the bank of swarm particles" (page 12) -> "the pool of swarm particles"
Author Response
We really appreciate all insightful comments and useful suggestions, which definitely will help us to improve the quality and readability of our manuscript. The manuscript has been revised according to the comments. In order to make the revising contents more clearly, the changes according to the reviewer#1 and reviewer#2 comments for the grammar, spelling and syntax problems will be highlighted in Blue; the changes according to the reviewer#3 comments will be highlighted in Yellow.
Reviewer #2
The paper addresses the problem of autonomous navigation of Unmanned Surface Vehicles (USVs) by enhancing conventional Particle Swarm Optimization (PSO) techniques. The proposed enhancements include: (i) Linear descend of inertia weight, (ii) Adaptive control of acceleration coefficients, and (iii) Random grouping inversion. These enhancements are combined into three algorithms: APSO (ii), AWPSO (i)+(ii) and AWIPSO (i)+(ii)+(iii). These algorithms are compared against conventional PSO in three metrics: trajectory length, CPU time, and swarm size.
The paper is well written (although there are some small issues - see below) and the presented enhancements of the conventional PSO are easy to understand. The experiments show that AWIPSO generates best quality solutions although taking more CPU time. On top of that, the approaches are validated in a real-world scenario. I don't have much to criticize in general and hence I suggest "minor revisions".
Detailed comments:
- "reducing the optimal path length" (Introduction) - Optimal path length means the shortest path length, hence it cannot be more reduced. Deleting the word "optimal" from that sentence will do the job.
- Tables 5 and 6 give a list of coordinates that (I believe) are depicted in Figures 9 and 10. As the tables are hard to read for humans, they can be removed.
- There are missing references (page 10 top) of Zhang et al. and Yan et al.
- "non-deterministic polynomial hard" (page 2) -> "non-deterministicaly polynomially hard" or better "NP-hard"
- Missing space before "In order to plan..." (page 4 bottom)
- "the bank of swarm particles" (page 12) -> "the pool of swarm particles"
Response: Thanks for the reviewer’s critical comment. All the suggestions are fully appreciated. In our opinion, all the comments are highly constructive and useful to revise our manuscript. Following the reviewer’s suggestion, we carefully checked the manuscript entirely ourselves to eliminate all the simple grammar errors such as the wrong usage of the verb conjugations and other sentences which are not easy to be understood. All the responses in this file are also carefully checked for smoothness. In addition, we have modified Tables 5 and 6 as Appendix Tables C1 and C2 since detailed information of points’ location could not be obtained accurately from Figures 9 and 10. Please checked all the highlighted changings in the revised manuscript.
Reviewer 3 Report
Review of Xin, Li, Sheng, Zhang, & Cui, "Multi-Sensor Fusion for Navigation of Unmanned Surface Vehicles Using Improved PSO"
The paper describes a particle swarm optimization method to address the traveling salesman problem for a small unmanned surface vessel (USV). Three strategies are proposed: linear descent, adaptively controlled acceleration, and randomly-grouped inversion.
The paper's title does not match the subject of the paper. "Multi-sensor fusion for navigation" implies that the paper is making a contribution in sensor fusion or Kalman filtering, but the USV uses a traditional GPS/INS for navigation. It appears the contribution of the paper is the application of PSO to the traveling salesman problem, not solving a sensor fusion problem. Nor is the traveling salesman problem attempting to solve an underlying sensor fusion problem (for example, where the waypoint locations are updated to reflect changes in the environment that must be detected): here the set of waypoints is defined a priori.
It is not completely clear that the authors' solution to the TSP is equal to or better than the known solutions. For example, the TSPLIB page states that the optimal solution to the rat99 problem is 1211; the best answer given by the authors (in Table 2) is 1248.12 (using the AWIPSO algorithm). How does solution time compare between the proposed algorithms and solutions in the literature?
I don't understand the distances given in Table 7. The shortest distance over the path is given as 12.44 meters, but the distance across the sailing site (e.g. from the lower left corner to the upper right corner of the graph in Figure 10a) is 300 meters. This is calculated using the script given here:
https://www.movable-type.co.uk/scripts/latlong.html
How can such a serpentine path be shorter than the distance across the site?
Author Response
We really appreciate all insightful comments and useful suggestions, which definitely will help us to improve the quality and readability of our manuscript. The manuscript has been revised according to the comments. In order to make the revising contents more clearly, the changes according to the reviewer#1 and reviewer#2 comments for the grammar, spelling and syntax problems will be highlighted in Blue; the changes according to the reviewer#3 comments will be highlighted in Yellow.
Reviewer #3
The paper describes a particle swarm optimization method to address the traveling salesman problem for a small unmanned surface vessel (USV). Three strategies are proposed: linear descent, adaptively controlled acceleration, and randomly-grouped inversion.
Response: Thanks for the reviewer’s critical comment. All the suggestions are fully appreciated. In our opinion, all the comments are highly constructive and useful to restructure our manuscript. We believe that the new or modified contents included in the revised version really have improved the quality of the manuscript. We hope all the modifications will fulfill the requirement to make the manuscript acceptable for publication in this archive journal.
Detailed comments:
Comment 1: The paper's title does not match the subject of the paper. "Multi-sensor fusion for navigation" implies that the paper is making a contribution in sensor fusion or Kalman filtering, but the USV uses a traditional GPS/INS for navigation. It appears the contribution of the paper is the application of PSO to the traveling salesman problem, not solving a sensor fusion problem. Nor is the traveling salesman problem attempting to solve an underlying sensor fusion problem (for example, where the waypoint locations are updated to reflect changes in the environment that must be detected): here the set of waypoints is defined a priori.
Response: Thanks for the reviewer’s critical comment. As the reviewer suggested, we have modified the title of the paper to “Application of Improved Particle Swarm Optimization for Navigation of Unmanned Surface Vehicles”. Please see details in Page 1, Line 1-2.
Comment 2: It is not completely clear that the authors' solution to the TSP is equal to or better than the known solutions. For example, the TSPLIB page states that the optimal solution to the rat99 problem is 1211; the best answer given by the authors (in Table 2) is 1248.12 (using the AWIPSO algorithm). How does solution time compare between the proposed algorithms and solutions in the literature?
Response: Thanks for the reviewer’s critical comment. We want to state that the solutions of our improved algorithm AWIPSO is comparable with the optimal solutions of TSPLIB. As the reviewer suggest, we added the known optimal solutions from TSPLIB page to Table 2, and more statement in the context to make our explanation clearer. In addition, we presented the comparative results of solution time in Section 3.1.3, Table 4. As is mentioned, all the improved algorithms require more time for computation than the CPSO under the same maximum number of iterations. It is necessary to extend the time-cost to a certain degree for the purposes of improving the precision of solution and avoiding being trapped into the local optimum. Moreover, further study will focus on decreasing the solution time for the task of real-time path planning. Please see details in Page 14, Line315, Page 15, Line 333-334, Page 19, Line 398-404.
Comment 3: I don't understand the distances given in Table 7. The shortest distance over the path is given as 12.44 meters, but the distance across the sailing site (e.g. from the lower left corner to the upper right corner of the graph in Figure 10a) is 300 meters. This is calculated using the script given here:
https://www.movable-type.co.uk/scripts/latlong.html
How can such a serpentine path be shorter than the distance across the site?
Response: Thanks for the reviewer’s correction. We have to admit that the path distance derived originally in Section 3.3.3 is wrong. We made a mistake when we converted the units from (Longitude, Latitude) to meters. We carefully recalculated and modified the values. Please see details in Page 27, Line 494, Table 5 (Table 7 in original manuscript).
Round 2
Reviewer 3 Report
The authors have addressed my technical concerns.
This manuscript is a resubmission of an earlier submission. The following is a list of the peer review reports and author responses from that submission.
Round 1
Reviewer 1 Report
The submitted paper considers a heuristic algorithm for the traveling salesperson problem which is a classical problem of combinatorial optimization. The subject of the paper is not in the scope of Sensors. An example of the application of the proposed algorithm to a problem which is formulated using the sensors terminology does not change the focus of the paper. Only one of 27 references is to a paper in a sensors related journal. The authors are advised to resubmit the manuscript to a journal on combinatorial optimization.
Reviewer 2 Report
The paper proposes improved PSO algorithms and applies them to solve the path navigation of multiple USVs. In general the paper is nicely written, and the results are supported by numerous comparative studies and simulations. The extensive, intuitive and numerical calculations have justified the performance of the algorithms.
My general comments are to encourage the authors to provide some theoretical justifications on why and how the proposed algorithms could outperform the conventional ones, i.e., why and how the three aspects "the linearly descending inertia weight, the adaptively controlled acceleration coefficients, and the randomly grouping inversion" could affect and improve the performance of the algorithms in certain optimization problems. Furthermore, how could one explore these aspects in some actual PSO algorithms (with quantitative analysis) to improve the convergence and optimization solutions?
The application to navigation of USVs looks like a separate problem, and I encourage the authors to carefully link the two problems in the current paper.
The English is generally acceptable but there are some typos and errors. The authors should carefully check the manuscript and further improve the writings.